# Quantification and parameterization of snowflake fall speeds in the atmospheric surface-layer

Spencer Donovan<sup>1</sup>, Dhiraj K. Singh<sup>1</sup>, Timothy J. Garrett<sup>2</sup>, and Eric R. Pardyjak<sup>1</sup>

<sup>1</sup>Department of Mechanical Engineering, University of Utah, Salt Lake City, UT, USA

Correspondence: Dhiraj K. Singh

(u6021818@utah.edu)

#### Abstract.

The modeled settling speed of frozen hydrometeors has important implications for the prediction of weather and climate. However, it is usually assumed, erroneously, that they fall in still air. Here, we present novel field measurements of individual snowflake microphysical properties and their settling velocities in atmospheric surface-layer turbulence. Individual snowflake motions are tracked in a laser light sheet using particle streak velocimetry (PSV). A hotplate device, the Differential Emissivity Imaging Disdrometer (DEID), is used to obtain precise estimates of snowflake mass, density, and size. Relative to calculated terminal velocities in still air, we present enhancements and reductions of snowflake settling speeds in turbulent air for a broad range of Reynolds and Stokes numbers. Functional forms describing actual snowflake fall speeds are presented and explored. In particular, a promising non-dimensional functional form for the ratio of actual particle fall speed to terminal velocity is presented in terms of turbulence intensity and a new variable called the shape density index or SDI, which is related to an individual hydrometeor's microphysical structure.

#### 1 Introduction

20

Predictions of precipitation amount, location, and duration are highly sensitive to parameterized expressions of how precipitation particles fall (Morrison et al., 2020). Substantial impacts on predictability have been identified for forecasts of hurricane trajectories (Fovell and Su, 2007), storm lifetimes (Garvert et al., 2005; Colle et al., 2005; Milbrandt et al., 2010), cumulative precipitation at regional scales (Colle et al., 2005; Milbrandt and Morrison, 2013; Morrison and Milbrandt, 2015; Hagos et al., 2018; Bao and Windmiller, 2021), precipitation extremes (Singh and O'Gorman, 2014), convective cloud dynamics (Parodi and Emanuel, 2009; Posselt et al., 2019), and weather and climate modeling (Jakob, 2002; Mitchell et al., 2008; Eidhammer et al., 2024).

Particle fall speeds play an important role in determining spatial and temporal variability of snow cover, the surface energy budget of the lower atmosphere, local hydrology, the mass balance of glaciers, and vegetation development (Cohen and Rind, 1991; Lehning et al., 2008). Existing models of snow-cover distribution, soil moisture, surface runoff, and river discharge use simple parameterizations of surface processes (Lehning et al., 2006). At smaller scales, topographical features modify the flow-field near the surface resulting in preferential deposition and spatial-temporal heterogeneity of snowfall distributions,

<sup>&</sup>lt;sup>2</sup>Department of Atmospheric Sciences, University of Utah, Salt Lake City, UT, USA

especially in mountain environments (Lehning et al., 2008; Mott and Lehning, 2010). Surface-layer turbulence is expected to be an important factor influencing the motion and deposition of frozen hydrometeors through their settling velocity.

In this paper, the ratio of the actual fall speed of a snow particle,  $V_p$ , to the calculated terminal velocity of the same snow particle under quiescent (no wind) conditions,  $V_t$ , is used to evaluate the degree to which the movement of the surrounding fluid modifies hydrometeor settling. Early experiments studying the effects of turbulence on particles falling in a gravitational field explored heavy spherical particles (Murray, 1970) and quantified the settling reduction using a normalized parameter we term here the settling enhancement ratio, or  $V_p/V_t$ . Solid spherical particles with known densities and terminal fall speeds in still fluid were compared to observed average fall speeds of identical particles falling through grid-generated turbulence, that is, turbulence produced by passing a steady flow through a mesh or grid to create nearly isotropic and homogeneous fluctuations. Murray (1970) conducted laboratory experiments using grid-generated turbulence and found that particle fall speeds were reduced by as much as 30% relative to their terminal velocities in still fluid.

30

50

By contrast, in a theoretical analysis, Maxey and Corrsin (1986) showed that inertial-particle settling speeds are generally enhanced in simulated flow fields over similar particles in a still fluid. This enhancement is owing to the "fast-tracking" of heavier particles that tend to sample the downward side of vortex boundaries. This effect was highlighted in subsequent publications that also showed instances of reductions in particle settling speed (Wang and Maxey, 1993a; Nielsen, 1993). Particles that are too fast or too large to be guided along the fast-track periphery of eddies, or if the vortices are short-lived, spend more time sampling upward-moving regions of the flow, resulting in "particle loitering" and a subsequent decrease in fall speed. When the eddy turnover time becomes comparable to the particle response time, however, even relatively large particles can momentarily adjust to the surrounding flow, leading to improved alignment with turbulent motion.

Good et al. (2014) highlighted that particles with high inertia and densities much greater than the fluid are increasingly unaffected by horizontal velocity fluctuations of the fluid, making them less likely to be fast-tracked, resulting in a reduction of the mean settling velocity. Nielsen (2007) identified a need for more experimental data due to the variety of flow structures with the same relative turbulence strength.

Within a random flow field and in the absence of particle inertia, Maxey (1987) showed the average particle velocity to be the same as the terminal velocity. In the case of very-high particle inertia, particles display ballistic trajectories decoupled from the fluid flow.

Generally, an increase or decrease in fall speed only occurs for particles within an intermediate range of inertia values, typically corresponding to Stokes numbers of about  $0.1 \le St \le 1$ , where the particle response time is comparable to the Kolmogorov time scale (Wang and Maxey, 1993b; Ireland et al., 2016). In a review of several published simulations and experimental results indicating enhancement and reduction in relative settling velocity, Nielsen (2007) noted the importance of particle density and inertia in the form of a Stokes number derived from the particle and fluid timescale ratio.

There is an increased likelihood of a particle having enough inertia to get into regions of fast-tracking with increasing Stokes number. That is, they showed loitering increasing as Stokes number decreases.

Numerical simulations by Fornari et al. (2016) of finite-sized particles with densities slightly greater than the fluid, show a reduction in settling velocity for Kolmogorov-scale particles. As  $V_t$  decreases and the streamwise turbulent velocity fluctuations

 $U'_x$  increase, a reduction in the particle settling speed occurs. A reduction in  $V_p$  is also observed for sub-Kolmogorov particles as  $V_t$  increases and  $U'_x$  decreases as particles are less likely to sample the regions of downdrafts, similar to the observations made by Nielsen (1993).

Fewer experiments exist that have measured the settling rate of snowflakes directly. The lack of experimental data is due to the difficulty of tracking and labeling frozen hydrometeors in their natural airborne state while simultaneously quantifying the microphysical attributes of each differing particle. The need to include the particle mass, density, and shape in fall speed calculations (Heymsfield and Westbrook, 2010) is highlighted by measurements obtained by Locatelli and Hobbs (1974) in still air that show that natural aggregates, even of similar size, can fall at very different terminal velocities ranging from about  $0.4 \text{ m s}^{-1}$  to  $1.2 \text{ m s}^{-1}$ .

Without measuring the inertial properties of snow, recent field experiments by Nemes et al. (2017) and Li et al. (2021b) observed a substantial increase in the settling velocity of tracked snow particles  $(V_p)$  in the atmospheric surface layer with average settling-rate enhancements of three and seven fold, respectively, over the corresponding still air terminal velocities. They define  $V_t$  in terms of a Stokes number timescale relationship as  $St = \frac{\tau_p}{\tau_\eta} = \frac{V_t/g}{\tau_\eta}$ , where  $\tau_\eta$  is the Kolmogorov time scale associated with the smallest turbulent eddies in the flow and  $\tau_p$  the aerodynamic response times. They determined values of  $\tau_p$  from the distribution of snowflake accelerations. The reported increase in settling speed was primarily attributed to surface-layer turbulence and preferential sweeping. Li et al. (2021a) observed the largest spread in velocity distributions associated with the narrowest spread in size distributions and visa-versa, suggesting that different densities could contribute to the counterintuitive findings.

In this paper, we measure Lagrangian settling velocities by tracking snowflakes over a wide range of snowfall conditions. Currently, limited measurements of the fall speed of frozen hydrometeors have been reported in the literature (Li et al., 2024a) that contain direct measurements of mass and density; most instead use an assumed aerodynamic density, which is an effective density inferred from a particle's drag behavior and depends on its shape, porosity, and orientation rather than its true material density. Our goal is to understand the impacts of turbulence on the fall speed as a function of the actual measured snowflake density and shape characteristics.

#### 2 Methodology

Atmospheric turbulence is generated by two primary components: wind shear and buoyancy driven by temperature gradients. Our focus is on the atmospheric boundary layer (ABL), specifically the surface layer, or the portion of the troposphere that is directly influenced by the surface on a timescale of approximately an hour or less and ranges in depth between hundreds of meters to a few kilometers (Stull, 1988). The turbulence data presented here were collected during the nighttime snow events when daytime buoyancy effects are weak. While diurnal behaviors are always present, during precipitation events, near-surface turbulence statistics are primarily determined by the wind's interaction with terrain complexities and vegetative-canopy structures. Turbulence statistics for each event are utilized to identify and segment cases into periods of high and low turbulence.

## 2.1 Field site

Field experiments were performed at the Mid-Collins (CLN) snow study plot in Alta, Utah, USA ( $40^{\circ}$  34′ 33.66″ N 111° 38′ 19.93″ W, elevation 2945 m) during the 2020-2021 winter season. The site averages 1300 cm of snowfall annually and has 17.4 days with at least 25 cm of snow per winter (Alcott and Steenburgh, 2010). The Alta study plot is located in a clearing surrounded by an  $\approx$ 10 m tall tree canopy surrounded by complex mountainous terrain. This site was chosen in-part to avoid the additional measurement of windblown snow that is typically lifted from exposed terrain features.

The instrumentation deployed for this work was collocated with existing operation instruments operated by Alta Ski Area as well as manual snow measurements conducted twice daily. Data collection at Alta consisted of one CSAT3 Campbell Scientific, Inc. sonic anemometer-thermometer that was maintained at  $\approx 1.5$  m above the snow surface that measured all three components of the wind and sonic temperature. These data were logged at 20 Hz to capture high frequency measurements of the along-wind component of the wind velocity  $u_x$ , the transverse component  $u_y$ , and the vertical velocity component  $u_z$ . One high-precision slow-response Vaisala HMP 155 thermometer/hygrometer was located at approximately 1.5 m above ground and sampled at 1 Hz. The CSAT3 and HMP 155 data were logged using a Campbell Scientific CR1000. A Differential Emissivity Imaging Disdrometer (DEID) (Singh et al., 2021b) was maintained at the same height as the sonic anemometer to measure individual snowflake mass, density, and geometric characteristics. Collocated with the DEID was a particle tracking system for measuring individual snowflake positions and velocities. A photograph of the main tower is presented in Fig. 1 (Figure. 1 was adapted from (Singh et al., 2023) with the permission of AIP Publishing. © 2023 AIP Publishing.). Detailed descriptions of the DEID and particle tracking system are given below.

In addition to the instruments on the main tower, a Multi-Angle Snowflake Camera (MASC) (Garrett et al., 2012) was deployed at  $\approx 1$  m above the snow surface on a mast located within 10 m of the main tower. The MASC captures high-resolution photographs of individual snowflakes; however, its data were not used in the present analysis. We have also used temperature data from Alta Ski Area's high-elevation Mt. Baldy observation station located at an elevation of 3373 m (40° 34' 3.72" N, 111° 38' 14.64" W)

Intensive Observation Periods (IOPs) were performed during most nighttime snowfall events throughout the 2020-2021 season and comprised millions of snowflake settling velocities and physical properties measurements. This paper focuses on four cases conducted during snow-event IOPs on 12 December 2020 (Case 1), 4 January 2021 (Case 2), 16 March 2021 (Case 3), and 26 March 2021 (Case 4). Details about the meteorology condition, observation time intervals, and microphysical parameters are provided in Table 2.

## 2.2 Snowflake imaging and tracking to measure $V_p$

The motions of settling individual snowflakes were imaged using a particle tracking system consisting of a laser sheet with a sampling volume (or region of interest, ROI) of 24.6 cm  $\times$  13.8 cm  $\times$  6.5 cm oriented normal to the viewing angle of a Nikon D850 SLR camera with a single focal length Nikon AF-S VR Micro - Nikkor 105 mm f/2.8G IF-ED lens. The SLR camera recorded 3840 pixel  $\times$  2160 pixel images with a spatial resolution of  $\approx$  64  $\mu$ m pixel<sup>-1</sup> at 30 fps within a vertical laser sheet

**Figure 1.** Field deployment installation overview at the Alta-Collins snow study plot including the DEID, PSV system, sonic anemometer, and hygrometer. The entire assembly is raised and lowered with a manual pulley system to maintain an approximately constant height above the snow surface throughout the winter. (This figure is adapted from (Singh et al., 2023) with the permission of AIP Publishing. © 2023 AIP Publishing).

created using three 10-W, 520 nm diode lasers and a collimator lens. The collimator lens creates a spread angle of  $\approx 6.8^{\circ}$ , which yielded a laser light sheet with a nearly constant thickness of  $\approx 6.5$  cm throughout the ROI. The camera arrangement and settings ensured that all particle images captured within the thickness of the laser sheet were well-focused (see Fig. 2). Recordings were performed at night so that only well-focused snowflake motions within the thickness of the light sheet were captured.

Snowflake tracking was performed using the Particle Streak Velocimetry (PSV) technique (Dimotakis and Koochesfahani, 1981) to measure Lagrangian velocities of individual particles  $V_p$  in the laser sheet for concentrations up to 3400 m<sup>-3</sup>. Here,  $V_p$  is the vertical component, which is positive when falling toward the ground. Typically, PSV is used for speed measurements, but we used the PSV technique to measure fall velocity using the fall angle and streamwise particle speed; details of the calculations are given in Appendix A.

**Figure 2.** Snowflake tracking system overview schematic. (1a) Three 10-W lasers form a light sheet located 1 m from the leading edge of the region of interest (ROI). (1b) SLR camera, 1 m normal to the center of the 18 cm by 15 cm ROI. (2a) Thermal camera, located ≈normal to the hotplate with a slight offset to avoid shadowing. (2b) Hotplate, with laser light sheet running through the thermal camera recording region. The bottom of the ROI was located 2 cm above the hotplate surface. (3a) CSAT3 sonic anemometer ≈1 m from the ROI.

Observed snowflakes moved both downward and upward depending on turbulence levels and the characteristic sizes of turbulent eddies (see Fig. A1). Positive downward and negative upward values of  $V_p$  are determined by segmenting the analysis into periods where the bulk particle motion or streamwise air flow is exclusively from left to right or vice-versa in the images as discussed in detail in Appendix A.

#### 140 2.3 Microphysical measurement of hydrometeors

Hydrometeors that exit the bottom of the laser viewing plane of the SLR camera are then captured by the hotplate surface of the DEID situated directed below. The DEID consists of an infrared camera pointed at a low-emissivity heated metal plate as described in detail in Singh et al. (2021a) and Rees et al. (2021b). The DEID makes use of the contrasting thermal emissivities of water ( $\varepsilon > 0.95$ ) and aluminum ( $\varepsilon < 0.1$ ) at the same temperature. A piece of Kapton tape ( $\varepsilon > 0.95$ ) is maintained on one side of the hotplate to provide a reference length scale as to provide the temperature of the plate. Images from the thermal camera were recorded with a resolution of 531 pixels  $\times$  362 pixels and a sampling rate of 15 Hz. Hydrometeors are imaged

within a 9 cm × 6 cm section of the hotplate located at its center appearing to the thermal camera due to the contrasting emissivity of water and metal imagery as bright images on a dark background. The hotplate is powered by a 120V, 5A supply and uses a digital proportional integral derivative (PID) feedback control mechanism to maintain a constant plate temperature. Binarized images were used to determine the contact area of each individual hydrometeor by counting the white pixels. The contact area of each snowflake was obtained from its melted imprint on the aluminum hotplate. As demonstrated by (Singh et al., 2021a), the difference between pre- and post-melting equivalent diameters is approximately 5%, indicating that the melt footprint reliably preserves the original two-dimensional geometry owing to the plate's near-90° contact angle.

150

160

165

180

The DEID provides accurate measurements of the mass of individual hydrometeors (m), along with the density  $(\rho_p)$ , the circumscribed ellipse area (A), and the actual contact areas  $(A_e)$  as described in Singh et al. (2021a, 2024) and illustrated in Fig. 3. Detailed calculations of A and  $A_e$ , along with the complexity, are provided in the following paragraph and in Appendix A, following the approach of Morrison et al. (2023). The circumscribed ellipse area (A) is obtained from an ellipse constructed using the bounding-box width and height, representing the smallest ellipse that fully encloses the projected shape of the hydrometeor.

From these parameters, individual snowflake-equivalent still-air terminal velocities can be estimated using an aerodynamic formula (Böhm, 1989; Heymsfield and Westbrook, 2010) namely,  $V_t = f(m, A_e, A)$ . Details of the calculation are provided in Singh et al. (2021a) and Singh et al. (2023). The estimated  $V_t$  using Heymsfield and Westbrook (2010) is  $\approx 15\%$  smaller than that calculated using Böhm (1989). Note that we estimated  $V_t$  using (Böhm, 1989) in this analysis, following the method described therein, with detailed calculation steps provided in Appendix B. . Also, size distributions are expressed here in terms of the effective diameter,  $D_{\rm eff}$ , which is the diameter of the equivalent circle of area  $A_e$  given by  $D_{\rm eff} = \sqrt{4/\pi A_e}$ .

The density of individual snowflakes is determined based a new method that exploits the rate of heat transfer during the melting of a hydrometeor on the hotplate (Singh et al., 2024). Combining microphysical measurements of the snowflakes with turbulence parameters of the surrounding air allows for the computation of frozen-particle Stokes numbers using the timescale ratio  $St = \tau_p/\tau_\eta$ . The turbulence Kolmogorov (micro)timescale,  $\tau_\eta$ , is given by  $\tau_\eta = (\nu/\epsilon)^{1/2}$  where  $\nu$  is the kinematic viscosity of air and  $\epsilon$  is the dissipation rate of turbulent kinetic energy. Here,  $\eta$  is the Kolmogorov micro(length)scale, which is the characteristic length scale of the smallest turbulent eddies in the flow (Tennekes et al., 1972).  $\tau_\eta$  is the timescale associated with these small-scale eddies and is calculated from 30-minutes of turbulence data, and individual particle response times are given by  $\tau_p = V_t/g$ . The density ratio parameter  $\beta$  (Mercado et al., 2012) is calculated for each snowflake as,

$$\beta = \frac{3\rho_f}{\rho_f + 2\rho_p}.\tag{1}$$

A value of  $\beta = 0$  corresponds to heavy particles that are perceived as having infinite inertia and  $\beta = 1$  implies particles that match the surrounding fluid density and behave as idealized fluid tracers. A density of air at approximately 2945 m above sea level of  $\rho_f = 0.9$  kg m<sup>-3</sup> is used here.

The combination of snowflake shape and density is characterized here with two newly introduced non-dimensional parameters we term the complexity and the shape-density index (SDI). Hydrometeor *complexity* is defined as the ratio of the area of a circumscribed ellipse, constructed using the bounding-box width and height, to the actual cross-sectional area  $(A_e)$  measured

Figure 3. (a) Illustration of the shape density index (SDI). Increasing SDI is shown in the figure. For a fixed actual area of a snowflake,  $A_e$ , at time  $\approx$ 0 (when the snowflake lands on the DEID hotplate), the density of the sampled snowflakes decreases, shown as melted water circles, resulting in a decreased equivalent area of the melted water,  $A_{\rm mel}$ , from the snowflake. (b) The Complexity of the snowflake is determined from  $A/A_e$  as used in Böhm (1989). The circumscribed ellipse area, A, is measured by the ellipse shown that completely encompasses the actual area,  $A_e$ , of the red colored snowflake.

on the DEID hotplate. The corresponding ellipse area is given by  $A = \pi ab$ , where a and b are the semi-axes derived from the bounding-box dimensions using MATLAB's Regionprops function, consistent with the definition used previously.

The complexity is then given by:

195

$$Complexity = \frac{\pi ab}{A_e},\tag{2}$$

where *a* and *b* are the semi-major and semi-minor axes of the ellipse, respectively. A perfectly circular particle has a complexity of unity, with higher values indicating increasingly irregular or elongated shapes.

The SDI accounts for the shape of the snowflake relative to a spherical water droplet, defined as

$$SDI = \frac{A_e}{A_{\text{mel}}}.$$
 (3)

where  $A_{mel}$  is the projected area of the equivalent spherical water droplet formed once the snowflake has melted, as given by

190 
$$A_{mel} = 1.2(m/\rho_w)^{2/3}$$
, (4)

where  $\rho_w$  is the density of liquid water and 1.2 is a dimensionless geometric constant. An SDI value of unity corresponds to a liquid water sphere. Large, aggregate-type snowflakes with low densities have high SDI. As illustrated in Fig. 3, small, dense snowflakes such as graupel have lower values of SDI (Morrison et al., 2023). Note that SDI is a dimensionless version of the specific surface area (SSA) per unit mass, which is widely used in the snow science literature (Fassnacht et al., 1999; Legagneux et al., 2002; Yamaguchi et al., 2019).

#### 2.4 Turbulence statistics

High-frequency wind velocity data were collected using a sonic anemometer situated at the same height and 1 m adjacent to the location of the DEID and SLR region of interest (ROI). To analyze the velocity data, the raw velocity signals are decomposed into a time averaged component (e.g.  $U_x$  and a fluctuating component  $u_x'$ ). Specifically,  $u_x = U_x + u_x'$ ,  $u_y = U_y + u_y'$ , and  $u_z = U_z + u_z'$  for the along-wind, transverse, and vertical velocity components of the wind velocity. To ensure that the anemometer is rotated into the streamwise coordinate system, two rotations are applied to each 30-minute averaging period. This procedure follows methods outlined by Wilczak et al. (2001) where the first rotation aligns the x and y-axis about the z-axis so that  $U_y = 0$ , aligning with the horizontal component of the flow. Calculation of turbulence parameters using sonic anemometer data is outlined in Appendix C. The integral time scale  $(\tau_L)$  and the integral length scale (L) are calculated using the temporal autocorrelation function of the horizontal-wind fluctuation. The turbulent Reynolds number  $R_\lambda = u'\lambda/\nu$ , where u' is the root mean square (r.m.s.) of along-wind velocity fluctuation and  $\lambda$  is the Taylor microscale length scale of turbulence. We compute the commonly used non-dimensional turbulence intensity TI from the sonic anemometry data as

$$TI = \frac{\frac{1}{2}(\sigma_{u_x}^2 + \sigma_{u_y}^2 + \sigma_{u_z}^2)^{1/2}}{U_x} = \frac{TKE^{1/2}}{U_x}.$$
 (5)

TKE is used in place of the streamwise velocity fluctuations  $u'_x$  to account for the chaotic multi-directional turbulence observed at the CLN field site.

## 3 Results and discussion

205

To quantify the broader impacts of turbulence and microphysical properties on snowflake fall speed, data collected from all IOPs are evaluated using different averaging periods. Averaging alleviates much of the synchronization discrepancies due the spatial separation of the three measurement systems (i.e., the DEID, sonic anemometer, and laser sheet). By averaging over longer periods, trends in the particle-turbulence interactions are more readily identifiable.

#### 3.1 Turbulence

Figure 4 shows a time series, using a 10-second moving average of the vertical wind velocity  $(u_z)$  and streamwise velocity  $(u_x)$ , that highlights differences in the mean and fluctuating velocity components during periods of relatively high turbulence and low turbulence associated with Cases 1 and 3, respectively. Turbulence statistics for each 30-minute case are summarized in Table 1 and are used to identify and segment cases into periods of high and low turbulence. For all cases, the winds are generally light (less than about 1 m s<sup>-1</sup>), however, clear differences can be seen between the turbulence characteristics of Cases 1 and 2 (high turbulence) versus Cases 3 and 4 (low turbulence).

**Figure 4.** 10-minute time series of rotated sonic anemometer data from the Case 1 high-turbulence dataset, indicated by the solid black lines, and the low-turbulence dataset from Case 3, indicated by the red dashed lines. A 10-second moving average is applied to the time series data. (a) Streamwise wind velocity. (b) Vertical wind velocity.

#### 3.2 Snowflake Characterization

Snowflake characteristics such as complexity and density are determinants of hydrometeor drag and hence the terminal fall speed (Böhm, 1989). Thus, we expect that understanding the microphysical characteristics of snowflakes is important for determining actual fall speeds. Figure 5 shows probability density functions or PDFs of effective diameter, mass, complexity, density, terminal fall speed, Stokes number, density ratio, SDI, and SSA obtained from DEID measurements of 86,086 individual snowflakes for a low-turbulence situation, Case 3. Datasets like these are unique and illustrate the wide range of values that microphysical variables take on during a single storm leading to large variability in fall speeds. The following are median values with corresponding lower and upper quartiles for the measured parameters:  $D_{\rm eff} = 1.4[1.1,1.8]$  mm; Mass = 0.06[0.03,0.15]mg; Complexity = 1.3[1.1,1.5];  $\rho = 54[41,72]$ kg m<sup>-3</sup>;  $V_t = 0.6[0.4,0.8]$ m s<sup>-1</sup>; St = 0.18[0.12,0.26];  $\beta = 0.02[0.02,0.03]$ ; SDI =8.5[6,12]; SSA = 0.03[0.01,0.04]. Furthermore, we also found that the mass distribution of the snowflakes follows a power-law function, as shown in Fig. 5b, departing from the more familiar negative exponential distribution observed for  $D_{\rm eff}$  (Gunn and Marshall, 1958). This discrepancy arises because our measurements directly capture particle

mass, whereas previous studies inferred it from mass–diameter relations in which the exponent varies between 1 and 3 (Rees et al., 2021a), leading to deviations from a pure power-law distribution.

Table 2 presents relationships that are also not typically available in the analysis of complex snow-storm events but are extremely useful for modeling. In particular, mass-diameter, density-diameter, and terminal settling speed-diameter parameterizations are shown for all four cases studied. While some of the parameterizations characterize the snowflakes well, most do not. This emphasizes the complexity of the distributions. Specifically, these distributions include different types of snowflakes which are expected to have their own relationships (Locatelli and Hobbs, 1974).

The distributions of snowflake effective diameters ( $D_{\rm eff}$ ) and particle densities ( $\rho_p$ ) under high and low turbulence conditions are shown in Fig. 6. High-turbulence cases (Case 1 and Case 2) exhibit broader  $D_{\rm eff}$  and  $\rho_p$  PDFs with slightly larger densities and almost identical mean sizes. These trends underscore the influence of turbulence on snowflake size and density variability. This may be a result of increased snowflake collisions that occur during higher turbulence (Grzegorczyk et al., 2023). These observations highlight the significant role of turbulence intensity in modulating snowflake microphysical properties.

The distributions of snowflake effective diameters ( $D_{\rm eff}$ ) and particle densities ( $\rho_p$ ) under high and low turbulence conditions are shown in Fig. 6. High-turbulence cases (Case 1 and Case 2) exhibit broader  $D_{\rm eff}$  and  $\rho_p$  PDFs with slightly larger densities and nearly identical mean sizes. This indicates that turbulence primarily affects the spread rather than the mean value of  $D_{\rm eff}$ . Increased turbulent fluctuations enhance relative motion, orientation variability, and collision or fragmentation frequency, leading to broader distributions even when the mean remains unchanged (Wang and Maxey, 1993a; Sundaram and Collins, 1997; Shaw, 2003; Grabowski and Wang, 2013).

## 3.3 Snowflake fall speeds

Direct measurements of individual snowflake settling velocities, obtained using PSV, and their terminal velocities  $V_t$  obtained from the DEID are presented in this section. Snowflake fall speeds,  $V_p$ , are compared to concurrent measurements of the snowflakes' terminal velocities  $V_t$  to identify settling reduction and enhancement periods and to understand the mechanisms governing the behavior.

Table 3 shows how fall-speed enhancement and reduction correlate with Stokes number for 30-minute averages. An enhancement in settling speed is noted when St < 1, while a reduction is observed when St > 1. Evidently, higher St values are associated with reductions in the average fall speeds  $V_p$  as snowflake trajectories become more horizontal. This is a rather surprising result that was discussed in Singh et al. (2023) and Garrett et al. (2025). It is important to note that in this case, the elevated  $St = \tau_p/\tau_\eta$  results from increased turbulence intensity, which reduces  $\tau_\eta$ , rather than from greater particle inertia or increased  $\tau_p$ . As illustrated in Fig. 7, individual particles with Stokes numbers less than 1 also tend to exhibit enhancement, while those with St > 1 show a reduction, similar to the 30-minute averages shown in Table 3. The extent of both effects is closely linked to the SDI. Figure 7a, representing a low-turbulence case, shows that particles with low St exhibit enhanced settling, with enhancement factors reaching up to 6 for particles with high SDI and values close to unity for particles with low SDI. Figure 7 also shows an inverse relationship between SDI and St.

Figure 5. PDFs of snowflake characteristics collected using the DEID, measured from a 6-hour continuous dataset comprised of 86,086 snowflakes at the CLN field site for Case 3. (a) Effective circular diameter  $D_{\text{eff}}$ , (b) Mass, (c) Complexity, (d) density  $(\rho_p)$ , (e) terminal velocity  $(V_t)$ , (f) Stokes number (St), (g) density ratio  $(\beta)$ , (h) shape density index (SDI), (i) specific surface area (SSA).

By looking at individual snowflakes and analyzing their orientation dynamics during free fall under varying turbulence conditions, our observations reveal that orientation is not fixed but evolves continuously as the particle descends (Garrett et al., 2025). Interestingly, the rate of change in orientation exhibits distinct statistical behaviors depending on the turbulence intensity. These orientation dynamics have direct implications for drag force estimation and terminal velocity prediction. Since drag is strongly dependent on the projected area normal to the direction of motion, variations in particle orientation lead to corresponding fluctuations in the instantaneous  $V_t$ . Specifically, we observe that terminal velocity estimates can vary significantly depending on whether the minimum or maximum projected area is used in the drag calculation. The maximum terminal velocity corresponds to the configuration with the smallest frontal area, while the minimum terminal velocity is associated with the largest area given with respect to the gravitational vector. Analysis of millions of individual snowflake measurements reveals that the ratio of maximum to minimum terminal velocity can be as large as 1.5. This result underscores the importance

Figure 6. PDFs of individual snowflake  $D_{\rm eff}$  and  $\rho_p$  data collected using the DEID. The dashed lines indicate the mean value for each dataset. PDFs of  $D_{\rm eff}$  are given in plots (a) and (b), and PDFs of  $\rho_p$  are given in plots (c) and (d). (a) and (c) are high-turbulence datasets for Case 1 and Case 2 while (b) and (d) are low-turbulence data sets for Case 3 and Case 4.

of accounting for orientation-induced drag variability when modeling particle settling, particularly for irregular, anisotropic particles such as snowflakes.

Figure 8, shows fall speeds as a function of particle density across a broad range of turbulence intensities. Both intuitively and formally (see Eq. B2 for the terminal velocity), we expect frozen hydrometeors with larger masses to fall at higher speeds in quiescent conditions. Within all 30-minute datasets, even for the minimum observed Reynolds number condition ( $Re_{\lambda}$  = 416), we see that turbulence effects are always present and this is in part why we see a decreasing  $V_p$  as density increases. Under high-turbulence conditions (Cases 1 and 2), settling velocities are significantly reduced, leading to longer atmospheric residence times. Notably, we also observed an increase in snowflake density under these conditions similar to the processes occurring within clouds, suggesting that prolonged suspension in turbulent flow may facilitate the formation of denser snowflakes (i.e., snowflakes with low SDI). These results are observational and emphasize the statistical relationships among turbulence intensity, particle density, and fall speed. A complete physical interpretation would require concurrent measurements of the local flow field and particle-scale forces, which are beyond the scope of this study.

Figure 7. 0.5-second averaged data showing the fall speed ratio relative to Stokes number, and colored by SDI. The maximum enhancement of  $V_p$  occurs as St approaches zero and SDI is high. Also, fall speeds of particles with low SDI are not as impacted by turbulence (a)Case 4 low turbulence dataset. (b) Case 1 high turbulence dataset.

**Figure 8.** 1-min averaged snowflake snowflake vertical fall speed as a function of snowflake density for all cases. Fall speeds were measured using PSV and densities were measured using the DEID.

Figure 9. One-minute averaged data from all four cases (120 minutes total) showing fall-speed enhancement  $(V_p/V_t)$  as a function of TKE. Marker color represents SDI.

Figure 9 shows the relationship between the  $V_p/V_t$  and TKE for all cases. Each data point represents a one-minute average collected over a 120-minute observational period. The data exhibit a clear inverse relationship, showing that increased turbulence intensity tends to reduce the fall speed of snowflakes. The observed suppression of fall speed at high TKE suggests that turbulent updrafts and eddies play a significant role in delaying particle settling. This is surprising since both experimentally and numerically, loitering is not expected, except in constrained channels. The observed inverse relationship between turbulence intensity and fall speed is unexpected since previous studies, generally at lower  $Re_{\lambda}$ , reported sweeping rather than loitering behavior Li et al. (2024b). Our results extend into higher  $Re_{\lambda}$  regimes and show that this inverse trend persists even in open, fully developed turbulence, indicating that settling dynamics depend on the turbulent scale and may not be fully captured by existing low- $Re_{\lambda}$  models. Furthermore, our measured loitering and sweeping is an order of magnitude larger than the results of Li et al. (2024b). Note that the scatter about the best fit line is largely due to variations in SDI.

Figure 10 shows PDFs of observed hydrometeor fall speeds  $V_p$  and terminal velocities  $V_t$  determined using DEID measurements and Eq. B2. During the high-turbulence intensity Case 1 and Case 2 events, many particle streaks had upward trajectories, 33.5% and 8.9%, respectively, representing a clear reduction in  $V_p$  compared to the terminal fall speed. A mean settling speed reduction of 57.1% was measured relative to the mean terminal velocity during a 30-minute high-turbulence event (Case 1) with  $Re_{\lambda} = 13187 \pm 923$ ; a mean reduction of 28.3% was observed during another high-turbulence event (Case 2) with  $Re_{\lambda} = 10163 \pm 711$  (Table 3 and Fig. 10a,b). Conversely, during periods of lower turbulence (Case 3 and Case 4) with

300

 $Re_{\lambda}=467\pm32$  and  $Re_{\lambda}=416\pm29$ , respectively, an enhancement in settling speed of 19.4% and 155.2% was observed (see Fig. 10c,d).

Figure 10. Probability density functions (PDFs) comparing actual vertical fall speed  $V_p$  from PSV measurements and terminal fall speed  $V_t$  derived from DEID measurements using Böhm (1989) throughout the four 30-minute cases. The dotted blue lines denote mean  $V_p$  and orange dashed lines mean  $V_t$  for each event with their associated distributions in the same color. 30-minute mean values are shown in the legends, and the mean turbulence intensity is also shown on the figures. The shaded regions indicate  $V_p$  values greater than the maximum  $V_t$  for each dataset, representing a small portion the total PDF (between 0.02% and 1.18% for all cases) (a) Case 1 dataset comprises 288,725 snow particle measurements. (b) Case 2 dataset comprises 687,893 snow particle measurements. (c) Case 3 dataset comprises 378,395 snow particle measurements (d) Case 4 dataset comprises 672,096 snow particle measurements.

The results from these cases point to a reduction in fall speed for the case of high turbulence and an enhancement for lower turbulence. Similar numerical and experimental studies (Wang and Maxey, 1993a; Nielsen, 1993; Good et al., 2014; Nemes

et al., 2017; Li et al., 2021c) primarily find enhancements in settling speed for small inertial particles as turbulence increases  $V_p$  by way of preferential sweeping mechanisms. What distinguishes the measurements presented here is the availability of the DEID to directly measure hydrometeor microphysical snowflake properties for computation of  $V_t$  and a more precise evaluation of the degree of settling reduction or enhancement.

Reductions in  $V_p$  during high-turbulence events are associated with broadening of the settling distribution with an increasing fraction of snow particles that move near horizontally or upwards. The lowest turbulence event with  $Re_{\lambda}$  = 416 and TI = 0.27 observed in Case 4 has a much narrower distribution than the other cases. Interestingly, the two-dimensional speed (in the plane of the laser sheet, see  $V_v$  in Appendix A) of the snowflakes is however comparatively similar among each of the four datasets.

Also notable in Fig. 10 is that, for all events, the fall-speed distributions have long tails implying high fall speeds occur more often than expected from calculated values of  $V_t$ . Secondly, fall-speed distributions during lower-turbulence events are bimodal with peaks near zero fall speed and at higher speeds. High-turbulence cases are unimodal with peaks near zero settling speed. Note that distributions of density and effective diameter during this period do not show any bimodal behavior (Fig. 6).

### 3.4 Power-Law fall-speed parameterization

By combining the critical variables collected by all components of the experimental setup (i.e., DEID, PSV, and sonic anemometer), we have shown that snowflake fall speed is a function of snowflake density, snowflake shape, and turbulence intensity. Expressed in terms of non-dimensional parameters, we hypothesize this functional form to be  $V_p/V_t = f(\text{SDI}, \text{TI})$ , with the specific form of the equation given by the following power-law

$$\frac{V_p}{V_4} = A_0 \text{SDI}^a \text{TI}^b. \tag{6}$$

Figure 11 shows this power-law along with data points from all of the cases studied, where the constants  $A_0 = 0.013$ , a = 1.65, and b = -0.32 were determined by multiple linear regression. The resulting  $R^2$  for the fit is 0.75. The data-point markers in Fig. 11 are colored by SDI and sized by TI. In these results, TI governs both the enhancement and reduction of  $V_p/V_t$ , consistent with the findings in Sections 3.2 and 3.3. Figure 11 illustrates the inherent coupling that is observed between TI and SDI (i.e., snowflake microphysical characteristics). That is, storms with high levels of turbulence are typically associated with low SDI values, while storms with low-turbulence levels are associated with high SDI values. Thus, we expect to see light, fluffy snowflakes during periods of light winds and low turbulence.

#### 335 4 Conclusions

We present direct measurements of individual snowflake microphysical properties and their fall velocities within surface-layer atmospheric turbulence, obtained using a sonic anemometer, a particle-tracking system, and DEID. The DEID directly measures snowflake size, mass, and density (Singh et al., 2021b; Li et al., 2023), enabling coupled characterization of microphysical and dynamical properties under natural turbulent conditions. Four snow events were observed at a high alpine location in Utah,

Figure 11. One-minute averaged data from all four cases (120 minutes total) showing the empirical fall-speed enhancement  $(V_p/V_t)$  parameterization. Marker color represents SDI and marker size scales with turbulence intensity (TI).

USA, two with relatively high turbulence and two with low turbulence. Measurements were made of the particle terminal velocity  $V_t$  using the DEID and the actual fall speed  $V_p$  using the particle tracking system to obtain settling enhancements  $V_p/V_t$ for comparison with turbulence levels and snowflake microphysical properties, including a new microphysical parameter called the shape-density index or SDI that accounts for snowflake density and complexity.

We found that the fall speed of frozen hydrometeors is a strong function of both turbulence intensity and SDI, and that SDI 345 and turbulence intensity seem to be coupled. Mean snowflake settling rates can be many times slower than terminal speeds in high turbulence and faster in low turbulence. Enhancements are greatest for low-density snowflakes with high SDI. A powerlaw functional form is proposed to describe particle settling enhancement as a function of turbulence and SDI. Including SDI in the fitting significantly improved the model performance, with the coefficient of determination (R<sup>2</sup>) increasing from 0.62 to 0.75. Specifically, the value for versus TKE alone was 0.62, while the combined fit with SDI and turbulence intensity (TI) yielded 0.75. This new parameterization has the potential to be implemented in numerical models that predict snowfall

amount, location, and duration, with some caution that a wider range of turbulence conditions should be considered and that the measurements described here were obtained in a clearing within a dense heterogeneous tree canopy with close proximity to mountainous alpine terrain.

### Appendix A: Particle Streak Velocimetry methodology

This appendix briefly describes our implementation of Particle Streak Velocimetry (PSV). PSV is a useful method for measuring flow fields filled with particles such as snowflakes. It quantifies two-dimensional Lagrangian velocities from particle trajectories captured by single-frame long-exposure images. Snow particle motion was captured within the region of interest (ROI), composed of a 24.6 cm  $\times$  13.8 cm  $\times$  6.5 cm volume. The captured snowflake images have 3840 pixels  $\times$  2160 pixels with a resolution of 0.06 mm per pixel. The camera arrangement and settings ensure that all particle images captured within the thickness of the laser sheet are well-focused. Recordings are performed at night so that only the well-focused snowflake motions within the thickness of the light sheet are captured. The Sobel Image edge detection method is applied to distinguish snowflakes from the background, which assumes a steep intensity gradient exists along the edges of the particle streak images. By locating the points where the local intensity gradient slope reaches maximum and minimum, a global threshold is applied to all captured images Vincent et al. (2009). In some instances, a pre-processing step is added to particle streak recordings, where the brightness threshold is adjusted using Adobe Premiere® (22.0) to remove lower brightness areas of condensed particle plumes. Snowflake streak image analysis is performed using the MATLAB® Image Processing Toolbox. A binary threshold of 25/255 = 0.10 assigns a value of unity for pixels containing frozen hydrometeors and zero value where no hydrometeors are present. The threshold may be attenuated to remove the occasional presence of image static that results from increased camera temperature during differing IOP conditions (See Fig. A1). Attenuating the binary threshold to determine the boundaries of the particle streak image may cause uncertainty in the aspect ratio of particle streaks resulting in inaccuracies in streak length and width measurement, and an associated of  $\pm 10\%$ .

The start and end points of streaks provide the extent of the bounding boxes for measuring the streak length and local azimuth angles, assuming that the Lagrangian trajectory of the snowflake within the exposed streak is linear. The individual hydrometer fall speeds are calculated by measuring the streak length, minus the particle dimension minimum width (perpendicular to streak length), of the exposed snowflake streaks relative to the 1/125 s frame exposure time. This concept is illustrated in Fig. A2. Streaks whose paths intersect with the frame edges of the ROI are removed. During high turbulence, when the turbulence Reynolds number,  $Re_{\lambda} \gtrsim 10000$ , we observe many particles moving with upward and downward velocities. During these periods, the fall angle resolves upward and downward motions (See Fig. A1).  $\pm V_p$  values are decoupled by segmenting the analysis into periods where the bulk particle motion is exclusively from left to right or vice-versa. In the images, the horizontal velocity of the streak is greater than vertical velocity. During periods of decreased turbulence,  $Re_{\lambda} \lesssim 400$ , we observe particle streaks with only downward velocity. We used this method for the first time, to determine both particle speed and direction simultaneously.

Figure A1. Example illustration using particle streak velocimetry (PSV) to determine streak edge and fall direction. (a) Binary streak image conversion from gray scale to black and white based on plotted pixel intensity measurement extending beyond both edges along the minor axis of the streak image. Pixels with an intensity below the determined threshold indicate the edges of the streak. (b) Upward and downward particle motion according to bulk observed particle horizontal trajectory and fall angle, where streaks with  $\theta \ge 90^\circ$  = upward movement  $-V_p$  and  $\theta 

**Figure A2.** PSV methodology for extracting velocity and fall angle of each snowflake trajectory from a single frame. Image frame was captured during the 4 January 2021 snow event with an exposure time of 1/125 s.

Figure A3. DEID installation schematic and imaging technique. (a) Schematic of the DEID. The top surface of a roughened heated aluminum plate imaged by a thermal camera is dark due to its low infrared emissivity. Hydrometeors with a high emissivity that reach a high temperature on the heated plate show as bright regions from which the hydrometeor's size and area can be measured by counting pixels. (b) Black and white binary thermal images of snowflakes in various stages of melting and evaporation on the DEID heated plate observed at Alta. The emissivity  $\varepsilon$  of snow and aluminum are noted. (c) Enlarged image illustrating the definitions of  $D_{\rm eff}$ ,  $D_{\rm min}$ ,  $D_{\rm max}$ . This figure is reproduced from (Singh et al., 2021b), published under the Creative Commons Attribution 4.0 License (CC BY 4.0).

#### Appendix B: Terminal velocity estimation from the DEID

385

390

Enhancements of the settlings speeds of frozen hydrometeors are quantified by comparison to the terminal fall velocity of the same precipitation particles in still air. Normally, a power-law relationship is used to solve for  $V_t$  of a snow particle, namely

$$V_t = \alpha D^{\beta}. \tag{B1}$$

Where D is the maximum dimension of the particle and  $\alpha$  and  $\beta$  are constants that have been determined empirically for a range of particle types Locatelli and Hobbs (1974). Based on fluid dynamical considerations, the terminal velocity  $V_t$  is calculated assuming non-linear drag forces based on formulae developed by Böhm (1989). Terminal velocity  $(v_t)$  is governed by the cross-sectional area normal to the flow. For each hydrometeor, multiple in-flight images are captured in the x-z plane (where the apparent area varies with rotation), along with one deposition footprint in the x-y plane on the hotplate. The footprint correlates with the maximum projected area (Singh et al., 2021a), and in still air, the mean orientation tends toward broadside,

with variance decreasing as turbulence weakens (Garrett et al., 2025). Accordingly, the x-y footprint is used as a proxy for the maximum cross-section to estimate  $v_t$ . The DEID instrument enables direct measurement of individual hydrometeor properties, including mass (m), circumscribed ellipse area (A), and contact area on the collection plate  $(A_e)$ . The circumscribed ellipse area (A) is determined from an ellipse constructed using the bounding-box width and height, representing the smallest ellipse that fully encloses the projected area  $(A_e)$ . (Singh et al., 2021a, 2024). Using these parameters, the still-air terminal velocity of individual snowflakes can be estimated via an aerodynamic formulation (Böhm, 1989)

$$V_t = \frac{Re\,\eta}{2\,\rho_a} \left(\frac{\pi}{A_e}\right)^{1/2}.\tag{B2}$$

Where,  $\rho_a$  and  $\eta$  are the density of air and the dynamic viscosity of air, respectively, and the particle Reynolds number Re is given by

$$Re = 8.5 \left[ \left( 1 + 0.1519 X^{1/2} \right)^{1/2} - 1 \right]^2.$$
 (B3)

Where the Davies number X is given by

$$X = C_D \cdot Re^2 = \frac{8 mg \,\rho_a}{\pi \,\eta^2} \,(\frac{A}{A_e})^{1/4}. \tag{B4}$$

where g is the gravitational acceleration.

A modification to Equation B3 has been proposed by Heymsfield and Westbrook (2010) to account for all types of natural ice particles, where

$$Re = 8.5 \left[ (1 + 0.48 X^{1/2})^{1/2} - 1 \right]^2.$$
 (B5)

Comparison of  $V_t$  calculations using the equations of Heymsfield and Westbrook (2010) and Böhm (1989) are shown in Figure B1. Generally, the Heymsfield and Westbrook (2010) estimates are lower. Considering a dataset (Dec 17 2020 from CLN field site) comprised of 85,000 snowflakes measured on 30-minutes duration, the average value of  $V_t$  for B89 is 0.59 m s<sup>-1</sup>) based on the Böhm (1989) parameterization and 0.49 m s<sup>-1</sup> based on the Heymsfield and Westbrook (2010) parameterization.

Figure B1. a) Comparison of terminal velocity  $V_t$  values calculated using parameterizations by Heymsfield and Westbrook (2010) (HW10) Böhm (1989) (B89). (b) Corresponding distributions.

## **Appendix C: Turbulence statistics calculations**

420

425

Using Taylor's hypothesis, turbulence length scales are derived from temporal sonic anemometry data (see e.g., Singh et al. (2023)). The largest turbulent structures are quantified using the integral time scale  $\tau_L$  and the integral length scale L. The integral scales are quantified by integrating the autocorrelation of the streamwise velocity fluctuations following Tennekes et al. (1972). The Taylor microscale length scale  $\lambda$  is determined by the point where a parabola, fit to the first three data points of the autocorrelation function, crosses the x-axis (Tennekes et al., 1972). Following the methods outlined in Stull (1988), the dissipation rate of TKE,  $\epsilon$ , is estimated from the inertial subrange of the 1D  $(u_x)$  energy density spectrum. Estimates of the energy dissipation rate are used to calculate  $\eta = (\nu^3/\epsilon)^{1/4}$  and  $\tau_{\eta} = (\nu/\epsilon)^{1/2}$  representing the Kolmogorov length scale and time scale respectively.  $\nu$  is the kinematic viscosity of air at the corresponding temperature during a given IOP. We used a 30-minute averaging period for turbulence and particle statistics, consistent with standard boundary-layer analyses (Kaimal & Finnigan, 1994; Stull, 1988; Shaw et al., 1998). This duration captures dominant large-eddy motions while avoiding mesoscale non-stationarity. Sensitivity tests using 60- and 120-minute windows showed 

Figure C1. 30-minute turbulence conditions measured at the CLN field site on 4 January 2021. (a) Autocorrelation of the streamwise velocity fluctuations. The integral of the portion of the signal that is correlated with itself is represented by the shaded area and measured by the integral I. The Taylor microscale is marked by  $\lambda$  located on the x-axis. (b) 1-D energy density spectrum, with the inertial subrange indicated by the portion with -5/3 slope. The region ahead of the inertial subrange shows where energy is being added to the flow by the largest turbulent structures and the related region after represents the smallest dynamically significant scales.

## Data availability

The data that support the findings of this study are available from the corresponding author upon reasonable request.

*Author contributions.* T.J.G. and E.R.P. conceived of the project. S.C.D. and D.K.S led the experiments and analysis of data in consultation with E.R.P and T.J.G.; All authors contributed to writing and editing the paper.

Competing interests. Conflict of interest: The DEID technology is protected through patent US20210172855A1 co-authored with D.K.S., E.R.P., and T.J.G. and is commercially available through Particle Flux Analytics, Inc. T.J.G. is a co-owner of Particle Flux Analytics, Inc. which has a license from the University of Utah to commercialize the DEID. Some authors are members of the editorial board of Atmospheric Chemistry and Physics.

Disclaimer: TEXT

Acknowledgements. We thank Allan Reaburn and his colleagues at Particle Flux Analytics, Inc. for their contributions to the development of the DEID, Dave Richards, Jonathan Morgan, and the Alta Ski Patrol for field support, as well as Travis Morrison for their contributions to the experimental setup in the field. This work was supported by the U.S. National Science Foundation through the following grants: PDM-1841870 and PDM-2210179)

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

Table 1. 30-minute turbulence statistics, collected from the sonic anemometer 1.5 m above the snow surface from the CLN site. See Appendix C for calculation details.  $U_x$  is the mean wind speed,  $\sigma_{u_z}$  is the standard deviation of the vertical velocity fluctuations,  $\tau_L$  is the integral length scale of turbulence, L is the Monin-Obukhov lengthscale,  $\epsilon$  is the dissipation rate of turbulent kinetic energy,  $\eta$  is the Kolmogorov lengthscale,  $\tau_{\eta}$ , is the is the Kolmogorov time scale,  $\lambda$  is the Taylor lengthscale,  $Re_{\lambda}$  is the Reynolds number based on the Taylor lengthscale, and TKE is turbulent kinetic energy.

| Case   | Period      | $U_x$        | $\sigma_{u_z}$ | $	au_L$ | L     | $\epsilon$                        | η    | $	au_n$ | λ     | $Re_{\lambda}$ | TKE            |
|--------|-------------|--------------|----------------|---------|-------|-----------------------------------|------|---------|-------|----------------|----------------|
|        | (MST)       | $(m s^{-1})$ | $(m s^{-1})$   | (s)     | (m)   | $(\mathrm{cm}^2~\mathrm{s}^{-3})$ | (mm) | (s)     | (mm)  |                | $(m^2 s^{-2})$ |
| Case 1 | 00:40-01:10 | 0.97         | 0.20           | 160.3   | 151.7 | 91.38                             | 0.87 | 0.045   | 351.9 | 13187          | 0.453          |
| Case 2 | 19:57-20:27 | 0.76         | 0.22           | 70.4    | 53.9  | 41.69                             | 1.05 | 0.066   | 341.7 | 10163          | 0.414          |
| Case 3 | 00:48-01:18 | 0.23         | 0.07           | 152.1   | 32.3  | 2.56                              | 2.56 | 0.420   | 67.3  | 467            | 0.019          |
| Case 4 | 23:57-00:27 | 0.19         | 0.03           | 262.3   | 42.8  | 3.85                              | 3.85 | 0.885   | 81.9  | 416            | 0.004          |

Table 2. Mass diameter, density-diameter, and terminal-velocity-diameter relations, as well as average Mt. Baldy site temperature  $(T_B)$ , average CLN site temperature  $(T_C)$ , relative humidity (RH), average wind speed (WS), and number of snowflakes measured by the DEID (N) are summarized for four cases. Mass is in milligrams (M), diameter  $(D_{\text{eff}})$  is in millimeters, density  $(\rho)$  is in kg m<sup>-3</sup>, and terminal velocity  $(V_t)$  is in m s<sup>-1</sup>.

| Case   | $M = aD_{\text{eff}}^b$ | $\rho = aD_{\rm eff}^b$ | $V_t = aD_{\mathrm{eff}}^b$ | $T_B$ | $T_C$ | RH  | WS           | N     |
|--------|-------------------------|-------------------------|-----------------------------|-------|-------|-----|--------------|-------|
|        | $[a, b, R^2]$           | $[a, b, R^2]$           | $[a, b, \mathbf{R}^2]$      | (°C)  | (°C)  | (%) | $(m s^{-1})$ | (#)   |
| Case 1 | [0.017, 2.77, 0.55]     | [76, -0.41, 0.19]       | [0.33, 0.83, 0.25]          | -16   | -12   | 71  | 0.99         | 1452  |
| Case 2 | [0.025, 2.69, 0.61]     | [75, -0.77, 0.40]       | [0.46, 0.64, 0.23]          | -6    | -3    | 92  | 0.80         | 8537  |
| Case 3 | [0.018, 2.80, 0.61]     | [64, -0.84, 0.40]       | [0.37, 0.68, 0.18]          | -7    | -5    | 95  | 0.23         | 12595 |
| Case 4 | [0.010, 2.74, 0.38]     | [54, -0.60, 0.16]       | [0.24, 0.81, 0.14]          | -11   | -9    | 92  | 0.22         | 8287  |

**Table 3.** 30-minute averaged snowflake fall speed and microphysical property data from the experiments conducted at CLN field site collected using the DEID and PSV during the four IOPs.

| Date   | $V_p$        | $V_t$        | $\frac{V_p}{V_t}$ | $V_h$        | $D_{eff}$ | $ ho_p$                   | SDI   | Complexity | St    | β     |
|--------|--------------|--------------|-------------------|--------------|-----------|---------------------------|-------|------------|-------|-------|
|        | $(m s^{-1})$ | $(m s^{-1})$ |                   | $(m s^{-1})$ | (mm)      | $({\rm kg}~{\rm m}^{-3})$ |       |            |       |       |
| Case 1 | 0.18         | 0.42         | 0.43              | 0.88         | 1.2       | 77.5                      | 11.3  | 1.37       | 1.07  | 0.021 |
| Case 2 | 0.48         | 0.67         | 0.72              | 0.88         | 1.5       | 68.0                      | 9.78  | 1.31       | 1.13  | 0.025 |
| Case 3 | 0.74         | 0.62         | 1.19              | 0.82         | 1.5       | 56.8                      | 11.18 | 1.36       | 0.226 | 0.031 |
| Case 4 | 0.74         | 0.29         | 2.55              | 0.76         | 1.1       | 55.3                      | 15.57 | 1.20       | 0.034 | 0.028 |