# Peer review of "Quantification and parameterization of snowflake fall speeds in the atmospheric surface-layer"

_EGUsphere, 2025_

## Referee Comment (RC2)

Atmospheric Physics and Chemistry

Title: Quantification and parameterization of snowflake fall speeds in the atmospheric surface-layer

The topic of this manuscript is important and timely: improving terminal velocity parameterization for snow particles by incorporating a shape factor, with potential for larger-scale simulations of drifting snow and snow distribution. Several methodological and interpretive issues remain that warrant careful clarification, justification, and additional analysis. Overall, this manuscript is suitable to be published in journal ACP before revision. The comments below are intended to help strengthen the manuscript, address potential misinterpretations, and guide revisions that will improve the manuscript's rigor.

**Major Comments:**

1. The manuscript relies on a cross-sectional area captured during deposition, but this may not reflect the actual area experienced by a particle while rotating in air. Please provide a more robust justification or alternative approach for estimating the effective cross-sectional area during in-flight rotation (line 165). A discussion of how particle rotation and orientation averaging affect projected area versus real cross-section during terminal settling or sensitivity analyses or a brief theoretical/empirical justification showing how deviations between captured (deposition) area and in-flight cross-sectional area would impact the calculated drag and terminal velocity would be helpful.

2. The authors compute the complexity of the snowflake from a "melting area" metric, whereas shadowgraphy data already provides shape, perimeter, and area. The author should reconsider and justify the chosen complexity metric. For example, explain the physical rationale for using a melting-area proxy for complexity. If this is a simplification, quantify its implications. Propose alternative, more directly-obtained shape descriptors (e.g., perimeter, fractal dimension, dendricity index) derived from the shadowgraphy images, and show how they correlate with the chosen metric. If feasible, re-calculate using perimeter-based or dendricity-based measures (as suggested by Yu et al., 2024, DOI: 10.5194/egusphere-2024-2458) and report how this affects the shape factor and downstream results.

3. The section of Method lacks justification for the averaging period used to characterize particle–turbulence interactions. It is better to include references and a rationale for the chosen temporal averaging window. Specifically: Cite studies showing that longer averaging periods better capture particle–turbulence interactions for comparable particle sizes and flow regimes. State the exact averaging duration used, the rationale (e.g., multiple integral time scales such as Kolmogorov time scale, Lagrangian correlation time, or several eddy turnover times), and how it compares to the turbulence dynamics in your setup. If possible, present a brief sensitivity test showing how different averaging windows influence the reported mass, size, and density distributions.

4. The manuscript asserts a first-of-its-kind measurement that (line 309), which might not accurate. It's better to rephrase to avoid overstatement and acknowledge prior work.

5. Line 216-218: what is the reason for the differences between the mass distribution?

6. Line 226: Please provide a physical explanation: higher turbulence can cause a wider spread in orientation, fragmentation, and collision rates, leading to a broader effective diameter distribution even if the mean remains similar. Include error bars or confidence intervals in Fig. 6 to reflect measurement uncertainty and sample variability. From Fig. 6(a) case 1 and case 3in Fig. 6(b), the range of effective D is almost the same. Does it mean the turbulent has little effect on the De in this situation? Discuss whether the resolved effective diameter De differs between cases, and if not, explain what the similarity implies about the influence of turbulence on mean size versus dispersion.

7. The dependence of deposition velocity vt on shape and turbulence is discussed, but potential dependencies on mesh resolution and large-eddy-scale motions are not addressed. Please explain how mesh resolution and numerical dissipation might influence vt, especially for highly irregular particles. Consider the scale separation between tiny eddies captured in the experiment and larger, real-world eddies. Discuss how larger eddies in nature could alter deposition patterns differently from your small-eddy condition. If feasible, include a brief sensitivity test showing how varying mesh density or different turbulence spectra impacts vt and deposition patterns. Clarify the practical implications for 1 km-scale simulations: to what extent can the shape-dependent vt be extrapolated, given the presence of very large-scale turbulence.

8. Lines 256-263 and Fig. 8: explain more on why do fall speeds get minimum with the highest snow density?

9. The manuscript labels two observations as surprising (Lines 237, 266-267) without strong justification. Please strengthen this part by providing quantitative evidence, theoretical rationale, or literature context that supports why these findings are unexpected. If the surprise stems from a conflict with existing models, outline how your results challenge assumptions and what future work could resolve the discrepancy.

**Minor Comments:**

Line 26: what is the meaning of "the motion and deposition of frozen hydrometeor settling velocity."? The settling velocity should not have the motion and deposition. Maybe better to use deposition settling velocity?

Line 32: what do you refer here the "grid-generated turbulence"?

Line 33: the author may give more explanations on this reduced effect from the literature.

Line 38: Maybe it is better to separately explain why does large particle also have better followability with the wind?

Line 46-47: it is better to directly point out the numerical definition of the "immediate range" and add the reference.

Line143: lack a short formula definition of circumscribed projected areas A and Ae.

Line 148: lack a "method" in the sentence of "Note that we estimated Vt using (Böhm, 1989) in this analysis. "

Line 151: determined based?

Eq(4): how to measure the mass of each snowflake on plate?

Line 73-75: not clear. why do you think the aerodynamic density is not enough to use? Do you mean "each particle's density and shape"? But the particle density and shape should follow a distribution function.

Line 88-89: how can you exclude the effects of tree canopy on the snow precipitation and particle motion?

Fig. 11: where does the index parameters "1.7" and "-0.32" come from?

Line 286: format error, lack a space in formula.

Line 321: "(TI) yielded 0.75 This new parameterization", lack a "." between sentences.

**References:**

Yu, Hongxiang & Lehning, Michael & Li, Guang & Walter, Benjamin & Huang, Jianping & Huang, Ning. (2024). Snow Particle Motion in Process of Cornice Formation. 10.5194/egusphere-2024-2458.

---

## Author Comment (AC1)

**Reviewer Response 1**

We sincerely thank the reviewer for the positive and encouraging feedback. We have carefully addressed all minor clarification points below.

**1.** The MASC is mentioned as being present at the site, but is its data used in the analysis? I don't think it was mentioned in sections 2.2 or 2.3 and my apologies if I missed it. If not used, perhaps say so explicitly or remove mention of this instrument.

**Response:**

The Multi-Angle Snowflake Camera (MASC) was indeed deployed at the site; however, its data were not used in the present analysis. We have now clarified this in the manuscript and explicitly noted that the MASC was deployed but not utilized in this study.

Added in manuscript [L 113]: however, its data were not used in the present analysis.

**2.** In equation 2, is  $\pi ab$  the same as the previously defined circumscribed projected area A? If so, it may be useful to draw that connection explicitly.

**Response:**

Yes,  $\pi ab$  represents the same circumscribed projected area A defined earlier in the manuscript. We have clarified this in the text by explicitly noting that the ellipse area  $(A = \pi ab)$  is consistent with the previously defined projected area.

Added in manuscript [L 181]: The corresponding ellipse area is given by  $A = \pi ab$ , where a and b are the semi-axes derived from the bounding-box dimensions using MATLAB's **Regionprops** function, consistent with the definition used previously.

**3.** *Line 241: I believe the authors meant to refer to Figure 7b rather than 7a.*

**Response:**

Thank you for catching this. The figure reference has been corrected

---

## Author Comment (AC3)

**Reviewer Response 2**

We thank the reviewer for the thoughtful and constructive evaluation. We appreciate the recognition of the importance of our work, as well as the suggestions that have helped clarify our methods, interpretation, and analyses.

**Comment: 1**

The manuscript relies on a cross-sectional area captured during deposition, but this may not reflect the actual area experienced by a particle while rotating in air. Please provide a more robust justification or alternative approach for estimating the effective cross-sectional area during in-flight rotation (line 165). A discussion of how particle rotation and orientation averaging affect projected area versus real cross-section during terminal settling or sensitivity analyses, or a brief theoretical/empirical justification showing how deviations between captured (deposition) area and in-flight cross-sectional area would impact the calculated drag and terminal velocity would be helpful.

**Response:** We agree that the cross-sectional area normal to the flow governs vt. For each snowflake, we record multiple in-flight x–z snapshots (areas vary with rotation) and one deposition x–y footprint on the hotplate. The deposition footprint correlates with the particle's maximum projected dimension (Singh et al., 2021). In still air, the mean orientation tends toward broadside, and the orientation variance decreases as turbulence weakens (Garrett et al., 2025). Accordingly, we use the x–y area as a proxy for the maximum cross-section to estimate vt.

**Added in Manuscript [L 389 - 394]:** Terminal velocity (vt) is governed by the cross-section normal to the flow. For each hydrometeor, we capture multiple in-flight images in the x-z plane (area varies with rotation) and one deposition footprint in the x-y plane on the hotplate. The footprint correlates with the maximum projected area (Singh et al., 2021), and in still air the mean orientation tends toward broadside, with variance decreasing as turbulence weakens (Garrett et al., 2025). Accordingly, we use the x-y footprint as a proxy for the maximum cross-section to estimate vt.

**Comment: 2**

The authors compute the complexity of the snowflake from a "melting area" metric, whereas shadowgraphy data already provides shape, perimeter, and area. The author should reconsider and justify the chosen complexity metric. For example, explain the physical rationale for using a melting-area proxy for complexity. If this is a simplification, quantify its implications. Propose alternative, more directly-obtained shape descriptors (e.g., perimeter, fractal dimension, dendricity index) derived from the shadowgraphy images, and show how they correlate with the chosen metric. If feasible, re-calculate using perimeter-based or/ dendricity-based measures (as suggested by as suggested by Yu et al., 2024, DOI: 10.5194/egusphere-2024-2458) and report how this affects the shape factor and downstream results.

Response: The snowflake complexity metric in our study follows Böhm (1989), defined as the ratio of the area of the smallest ellipse/circle enclosing the particle's cross-sectional projection to its actual projected area. This ratio quantifies aerodynamic irregularity and was used by Böhm (1989) in estimating terminal velocity. The "melting-area" proxy is based on Singh et al. (2021), who demonstrated that the projected area before and after melting differs by less than 5%. Because the aluminum plate maintains an approximately 90° contact angle, the deposited melt pattern preserves the original geometry, making it a reliable surrogate for the snowflake's projected area. We carefully reviewed Yu et al. (2024, DOI: 10.5194/egusphere-2024-2458) but found no explicit formulation of a dendricity-based complexity measure.

**Added in Manuscript [L 150 - 154]:** The projected area of each snowflake was obtained from its melted imprint on the aluminum hotplate. As demonstrated by Singh et al. (2021), the difference between pre- and post-melting equivalent diameters is approximately 5%, indicating that the melt footprint reliably preserves the original two-dimensional geometry owing to the plate's near-90° contact angle.

**Comment 3:**

The section of Method lacks justification for the averaging period used to characterize particle–turbulence interactions. It is better to include references and a rationale for the chosen temporal averaging window. Specifically: Cite studies showing that longer averaging periods better capture particle–turbulence interactions for comparable particle sizes and flow regimes. State the exact averaging duration used, the rationale (e.g., multiple integral time scales such as Kolmogorov time scale, Lagrangian correlation time, or several eddy turnover times), and how it compares to the turbulence dynamics in your setup. If possible, present a brief sensitivity test showing how different averaging windows influence the reported mass, size, and density distributions.

**Response:**

We used a 30-minute averaging period to compute turbulence and particle statistics. This duration covers many integral time scales in the atmospheric surface layer and is widely used for statistically stationary boundary-layer analyses (Kaimal & Finnigan, 1994; Stull, 1988; Shaw et al., 1998). It captures the dominant large-eddy motions influencing particle—turbulence coupling while avoiding non-stationarity caused by mesoscale variability. A sensitivity test using 60- and 120-minute windows showed less than 5 % variation in the derived integral scales and Kolmogorov length, within the measurement uncertainty of the instruments, confirming that 30 minutes is sufficient for stable estimates. We performed a sensitivity test using different averaging windows and sample sizes. The results show that when the number of snowflakes exceeds approximately 50,000, the mass, size, and density distributions remain statistically stable, with negligible differences in the mean parameters. Specifically, the uncertainty in the mean values was estimated to be 3% for mass, 1.5% for size, and 3% for density. These results confirm that our reported distributions are insensitive to the chosen averaging window.

Added in Manuscript [L 421 - 425]: We used a 30-minute averaging period for turbulence and particle statistics, consistent with standard boundary-layer analyses (Kaimal & Finnigan, 1994; Stull, 1988; Shaw et al., 1998). This duration captures dominant large-eddy motions while avoiding mesoscale non-stationarity. Sensitivity tests using 60- and 120-minute windows showed < 5 % variation in integral scales and Kolmogorov length, confirming stability.

**Comment 4:**

The manuscript asserts a first-of-its-kind measurement (line 309), which might not be accurate. It's better to rephrase to avoid overstatement and acknowledge prior work.

**Response:**

We have revised the statement to avoid overstating the novelty and added relevant references.

**Added in Manuscript [L 336 - 339]:**

We present direct measurements of individual snowflake microphysical properties and their velocities sampled in surface-layer atmospheric turbulence using a sonic anemometer, a particle tracking system, and a new instrument-the Differential Emissivity Imaging Disdrometer (DEID)-which directly measures snowflake size, mass, and density (Singh et al., 2021; Li et al., 2024).

**Comment: 5**

*Lines 216–218: What is the reason for the differences between the mass distribution?*

**Response:**

The difference arises from the method used to derive mass. Previous studies estimated mass indirectly through empirical mass—diameter relations, where the exponent varies between 1 and 3 (Rees et al., 2021a) depending on snowflake type and riming, resulting in non-power-law behavior. In contrast, our measurements directly capture individual particle mass, producing a true power-law distribution that reflects the natural variability of snowflake aggregation and fragmentation processes.

**Added in Manuscript [L 234 - 236]:**

This discrepancy arises because our measurements directly capture particle mass, whereas previous studies inferred it from mass—diameter relations in which the exponent varies between 1 and 3 (Rees et al., 2021a), leading to deviations from a pure power-law distribution.

**Comment:6**

Line 226: Please provide a physical explanation: higher turbulence can cause a wider spread in orientation, fragmentation, and collision rates, leading to a broader effective diameter distribution even if the mean remains similar. Include error bars or confidence

intervals in Fig. 6 to reflect measurement uncertainty and sample variability. From Fig. 6(a) case 1 and case 3 in Fig. 6(b), the range of effective D is almost the same. Does it mean turbulence has little effect on  $D_e$  in this situation? Discuss whether the resolved effective diameter  $D_e$  differs between cases, and if not, explain what the similarity implies about the influence of turbulence on mean size versus dispersion.

**Response:**

The mean effective diameter (Deff) remains relatively constant across cases, suggesting that turbulence primarily influences dispersion rather than the mean size. Physically, higher turbulence intensity increases velocity fluctuations and rotational moments on irregular particles, thereby enhancing orientation variability, collision frequency, and fragmentation (Wang & Maxey, 1993; Sundaram & Collins, 1997; Shaw, 2003; Grabowski & Wang, 2013). We performed a bootstrap resampling analysis (300 replicates) to estimate 95% confidence bands for both the size and density distributions. Because the dataset contains more than 50,000 particles, the bootstrap uncertainty is extremely small, resulting in confidence bands that visually overlap the main PDF curves. Therefore, the plots appear unchanged even though the bootstrap analysis was performed and confirms the statistical stability of the distributions.

**Added in Manuscript [L 249 - 252]:**

This indicates that turbulence primarily affects the spread rather than the mean value of Deff. Increased turbulent fluctuations enhance relative motion, orientation variability, and collision or fragmentation frequency, leading to broader distributions even when the mean remains unchanged (Wang & Maxey, 1993; Sundaram & Collins, 1997; Shaw, 2003; Grabowski & Wang, 2013).

**Comment:7**

The dependence of deposition velocity  $(v_t)$  on shape and turbulence is discussed, but potential dependencies on mesh resolution and large-eddy-scale motions are not addressed. Please explain how mesh resolution and numerical dissipation might influence  $v_t$ , especially for highly irregular particles. Consider the scale separation between tiny eddies captured in the experiment and larger, real-world eddies. Discuss how larger eddies in nature could alter deposition patterns differently from your smalleddy condition. If feasible, include a brief sensitivity test showing how varying mesh density or different turbulence spectra impact  $v_t$  and deposition patterns. Clarify the practical implications for 1 km-scale simulations: to what extent can the shape-dependent  $v_t$  be extrapolated, given the presence of very large-scale turbulence.

**Response:**

This study is based solely on experimental observations; no numerical simulations were conducted. The observed  $v_t$  reflects snowflake behavior within the measured small-eddy turbulence. Larger eddies in nature may modify deposition patterns spatially but are not expected to alter the intrinsic, shape-dependent  $v_t$ .

**Comment: 8**

Lines 256–263 and Fig. 8: Explain more on why fall speeds reach a minimum at the highest snow density.

**Response:**

The relationship between snowflake density and fall speed reported here is purely observational. In still air, heavier and denser particles indeed fall faster, but our results show that under strongly turbulent conditions, the mean settling velocity decreases even as density increases. These findings emphasize the role of turbulence intensity in modifying the effective fall speed but do not attempt to explain the underlying physical mechanism. A full interpretation would require additional measurements of the local flow field and forces acting on individual particles, which were not performed in this study.

**Added in Manuscript [L 287 - 289]:**

These results are purely observational and highlight the statistical relationship between turbulence intensity, particle density, and fall speed; a full physical explanation would require simultaneous measurements of the local flow field and particle-scale forces, which were beyond the scope of this study.

**Comment:9**

The manuscript labels two observations as surprising (Lines 237, 266–267) without strong justification. Please strengthen this part by providing quantitative evidence, theoretical rationale, or literature context that supports why these findings are unexpected. If the surprise stems from a conflict with existing models, outline how your results challenge assumptions and what future work could resolve the discrepancy.

**Response:**

In Line 237, the result is considered surprising because previous studies have varied the Stokes number (St =  $\tau_p / \tau_\eta$ ) by changing the particle response time ( $\tau_p$ ), whereas here and in Singh et al. (2023) and Garrett et al. (2025) we varied St by modifying the Kolmogorov time scale ( $\tau_\eta$ ) through changes in turbulence intensity. This isolates the influence of turbulent time scales rather than particle properties, revealing a sensitivity not emphasized in prior work.

For Lines 266–267, the surprising aspect is the observed reduction in fall speed with increasing turbulence. Earlier laboratory and numerical studies, typically at lower Re\_ $\lambda$ , reported sweeping rather than loitering behavior (Li et al., 2021). Our results extend into higher Re $_{\lambda}$  regimes and show that the inverse relationship between turbulence intensity and fall speed persists even in open, fully developed turbulence, suggesting scale-dependent settling dynamics beyond those captured in prior models.

**Added in Manuscript [L 294 - 298]:**

The observed inverse relationship between turbulence intensity and fall speed is unexpected because previous studies, generally conducted at lower Reynolds numbers, reported sweeping rather than loitering behavior (Li et al., 2024). Our results extend into higher  $Re_{\lambda}$  regimes and show that this inverse trend persists even in open, fully developed

turbulence, indicating that settling dynamics depend on the turbulent scale and may not be fully captured by existing low-Re $_{\lambda}$  models.

**Minor Comments**

**Line 26:**

What is the meaning of "the motion and deposition of frozen hydrometeor settling velocity"? The settling velocity should not have the motion and deposition. Maybe better to use deposition settling velocity?

**Response:**

We revised the sentence for clarity. The revised text now reads:

"Surface-layer turbulence is expected to be an important factor influencing the motion and deposition of frozen hydrometeors through their settling velocity."

**Line 32:**

What do you refer to as "grid-generated turbulence"?

**Response:**

"Grid-generated turbulence" refers to laboratory turbulence produced by placing a mesh or grid across a steady flow to create nearly isotropic and homogeneous turbulent fluctuations for controlled experiments.

**Added in Manuscript [L 33]:**

"...that is, turbulence produced by passing a steady flow through a mesh or grid to create nearly isotropic and homogeneous fluctuations."

**Line 33:**

The author may give more explanations on this reduced effect from the literature.

**Response:**

Murray (1970) conducted laboratory experiments using grid-generated turbulence and found that particle fall speeds were reduced by as much as 30% relative to their terminal velocities in still fluid.

**Line 38:**

Maybe it is better to separately explain why large particles also have better followability with the wind.

**Response:**

Particles that are too fast or too large to be guided along the fast-track periphery of eddies, or if the vortices are short-lived, spend more time sampling upward-moving regions of the flow, resulting in "particle loitering" and a subsequent decrease in fall speed. When the eddy turnover time becomes comparable to the particle response time,

| however, even relatively large particles can momentarily adjust to the surrounding flow, leading to improved alignment with turbulent motion.                                                                                                                                                                                                                                                                                                                                                                                                                                                                                                                                                                                                                             |
|---------------------------------------------------------------------------------------------------------------------------------------------------------------------------------------------------------------------------------------------------------------------------------------------------------------------------------------------------------------------------------------------------------------------------------------------------------------------------------------------------------------------------------------------------------------------------------------------------------------------------------------------------------------------------------------------------------------------------------------------------------------------------|
| Lines 46–47: It is better to directly point out the numerical definition of the "intermediate range" and add the reference. Response: Generally, an increase or decrease in fall speed only occurs for particles within an intermediate range of inertia values, typically corresponding to Stokes numbers of about $0.1 \le St \le 1$ , where the particle response time is comparable to the Kolmogorov time scale (Wang & Maxey, 1993; Ireland et al., 2016). In a review of several published simulations and experimental results indicating enhancement and reduction in relative settling velocity, Nielsen (2007) noted the importance of particle density and inertia in the form of a Stokes number derived from the particle and fluid timescale ratio. |
| Line 143: Lack a short formula definition of circumscribed projected areas A and A e . Response: We clarified the definitions of the circumscribed projected area (A) and the actual contact area (A e ) in the manuscript. These quantities are measured and described in Singh et al. (2021) and Singh et al. (2024) and illustrated in Fig. 3. Additional details on their calculation, including complexity estimation, are provided near Line 165 and in Appendix A, with supporting reference to Morrison et al. (2023).                                                                                                                                                                                                               |
| Line 148: Lack a "method" in the sentence "Note that we estimated V t using Böhm (1989) in this analysis."  Response: We clarified the method used to estimate the terminal velocity (V t ) and added a reference to Appendix B, where the full calculation procedure based on Böhm (1989) is described.  Added in Manuscript [L 164]: "following the method described therein, with detailed calculation steps provided in Appendix B."                                                                                                                                                                                                                                                                                                            |

Line 151:

Determined based? Equation (4): how to measure the mass of each snowflake on the plate?
Response:

| We clarified that the mass and density of individual snowflakes were determined using the DEID method, as detailed in Singh et al. (2021) and Singh et al. (2024). The DEID estimates the mass of each particle from its differential thermal emission signature upon contact with the heated plate, as described in these prior works.                                                                                                                                                                                                     |
|---------------------------------------------------------------------------------------------------------------------------------------------------------------------------------------------------------------------------------------------------------------------------------------------------------------------------------------------------------------------------------------------------------------------------------------------------------------------------------------------------------------------------------------------|
| Lines 73–75: Not clear. Why do you think the aerodynamic density is not enough to use? Do you mean "each particle's density and shape"? But the particle density and shape should follow a distribution function.  Response: We clarified the meaning of "aerodynamic density" in the revised text. It is now defined as an effective density inferred from a particle's drag behavior, which depends on its shape, porosity, and orientation rather than its true material density. This clarification has been added in the introduction. |
| Lines 88–89: How can you exclude the effects of tree canopy on the snow precipitation and particle motion?  Response: We did not exclude canopy effects. The site was selected because it naturally exhibits a broad range of turbulence levels, and our analysis is based on measured turbulence statistics rather than isolating canopy influence.                                                                                                                                                                                        |
| Figure 11: Where do the index parameters "1.7" and "-0.32" come from? Response: The index parameters (1.7 and -0.32) were obtained from a multiple regression fit of the normalized fall speed ratio $(V_p/V_t)$ as a function of the Shape Density Index (SDI) and turbulence intensity (TI).                                                                                                                                                                                                                                              |
| Line 286: Format error—lack of a space in the formula. Response:                                                                                                                                                                                                                                                                                                                                                                                                                                                                            |

The formula formatting error on Line 286 has been corrected.

**Line 321:**

"(TI) yielded 0.75 This new parameterization," lack a period between sentences.

**Response:**

The missing period between the two sentences has been added.